# Regulation of lung cancer initiation and progression by the stem cell determinant Musashi

**Alison G Barber**[1,2], **Cynthia M Quintero**[1,2,3,4], **Michael Hamilton**[1,2], **Nirakar Rajbhandari**[1,2], **Roman Sasik**[5], **Yan Zhang**[6], **Carla Kim**[7,8,9], **Hatim Husain**[2], **Xin Sun**[6], **Tannishtha Reya**[1,2,3,4]*

[1]Department of Pharmacology and Medicine, University of California San Diego School of Medicine, La Jolla, United States; [2]Moores Cancer Center, University of California San Diego School of Medicine, La Jolla, United States; [3]Herbert Irving Comprehensive Cancer Center, Columbia University Medical Center, New York, United States; [4]Department of Physiology and Cellular Biophysics, Columbia University Medical Center, New York, United States; [5]Center for Computational Biology and Bioinformatics, University of California San Diego School of Medicine, La Jolla, United States; [6]Department of Pediatrics, University of California, San Diego, La Jolla, United States; [7]Stem Cell Program, Division of Hematology/Oncology and Division of Respiratory Disease, Boston Children's Hospital, Boston, United States; [8]Department of Genetics, Harvard Medical School, Boston, United States; [9]Harvard Stem Cell Institute, Cambridge, United States

*For correspondence:
tr2726@cumc.columbia.edu

**Competing interest:** The authors declare that no competing interests exist.

## eLife assessment

This **important** study shows a significant role for Musashi-2 (Msi2) in lung adenocarcinoma. The authors provided **solid** data that support the requirement for Msi2 in tumor growth and progression, although the study would have been strengthened by including more patient samples and additional evidence regarding Msi2+ cells being more responsive to transformation. These findings are of interest to both the lung cancer and the RNA binding protein fields.

**Abstract** Despite advances in therapeutic approaches, lung cancer remains the leading cause of cancer-related deaths. To understand the molecular programs underlying lung cancer initiation and maintenance, we focused on stem cell programs that are normally extinguished with differentiation but can be reactivated during oncogenesis. Here, we have used extensive genetic modeling and patient-derived xenografts (PDXs) to identify a dual role for Msi2: as a signal that acts initially to sensitize cells to transformation, and subsequently to drive tumor propagation. Using Msi reporter mice, we found that Msi2-expressing cells were marked by a pro-oncogenic landscape and a preferential ability to respond to Ras and p53 mutations. Consistent with this, genetic deletion of *Msi2* in an autochthonous Ras/p53-driven lung cancer model resulted in a marked reduction of tumor burden, delayed progression, and a doubling of median survival. Additionally, this dependency was conserved in human disease as inhibition of Msi2 impaired tumor growth in PDXs. Mechanistically, Msi2 triggered a broad range of pathways critical for tumor growth, including several novel effectors of lung adenocarcinoma. Collectively, these findings reveal a critical role for Msi2 in aggressive lung adenocarcinoma, lend new insight into the biology of this disease, and identify potential new therapeutic targets.

## Introduction

Lung cancer is the second most common type of cancer. While the death rate due to lung cancer has decreased over time due to a drop in the prevalence of smoking, it is still by far the leading cause of cancer-related deaths in men and women, resulting in more deaths than colon, breast, and prostate cancer combined (*American Cancer Society, 2019*). Non-small-cell lung cancer (NSCLC) is the most prevalent form of lung cancer, accounting for 84% of lung cancer cases. Within this cancer type, adenocarcinoma is the most frequent subtype, accounting for 50% of all NSCLCs (*Davidson et al., 2013*; *Langer et al., 2010*). Most cases of lung cancer are diagnosed at late-stage—when there is regional or distant spread of the disease—at which time treatment options are limited (*Heist and Engelman, 2012*). Despite advances in lung cancer therapy, the 5-year survival rates for late-stage (i.e. regional or distant) NSCLC are 37–9%, respectively (*American Cancer Society, 2024*). These survival rates highlight the critical need for improved understanding of the biology of this disease in order to develop more effective treatments.

To understand the key dependencies of lung cancer, we have focused on stem cell programs which are often co-opted to perpetuate an aggressive, undifferentiated state. The stem cell fate determinant Musashi-2 (Msi2) is one such signal that has been shown to be critical in development as well as advanced cancers (*Fox et al., 2016*; *Ito et al., 2010*; *Koechlein et al., 2016*); however, its role in lung cancer is not well understood. Msi2 has been reported to be overexpressed in human lung adenocarcinoma and regional metastases, and its inhibition in human cell lines has been shown to reduce invasive and metastatic potential in vitro (*Kudinov et al., 2016*; *Moreira et al., 2010*). However, this study focused on in vitro analysis utilizing human NSCLC cell lines; thus, no evidence for a role of Msi2 in vivo using definitive genetic mouse models exist. Furthermore, whether Msi2 is a dependency for tumor initiation or required for continued propagation, and which pathways are regulated by Msi2 to drive tumor growth, also remain unknown. Here, we have utilized a combination of genetic models and patient-derived xenografts (PDXs) to determine the role of Msi2 signaling in the growth and progression of lung adenocarcinoma. Using Msi2 knock-in reporter mice and Msi2 conditional knockout mice, we show that Msi2 plays a dual role in lung adenocarcinoma, acting as a signal that not only is essential for initiation, but one that continues to be required post-establishment in both genetically engineered mouse models and in PDXs. Finally, using RNA-sequencing (RNAseq) analysis we show that Msi2 can trigger a range of pathways critical for tumor growth, including several new effectors of lung adenocarcinoma. These findings suggest that Msi2 is a critical regulator of lung adenocarcinoma and offer new insight into the signals that facilitate transformation and support disease progression.

## Results

### Msi2 is expressed in normal lung and in adenocarcinoma

To determine whether Musashi regulates the growth and progression of aggressive lung cancer we first determined whether Msi2 is expressed in stem cells of the distal lung. We used a $Msi2^{eGFP/+}$ knock-in reporter mouse to track endogenous Msi2 expression (*Fox et al., 2016*), and found that Msi2 is expressed in 39% of Lin⁻EpCAM⁺ lung epithelial cells (*Figure 1A*) and, more specifically, in 37% of cells enriched for Club/BASC cells and 26% of cells enriched for AT2 cells—the known stem cell populations of the distal lung (*Figure 1B, C*). Interestingly, although Msi2 expression could only be detected in a fraction of distal lung stem cells, widespread expression of Msi2 was often observed in lung adenocarcinomas formed in $Kras^{G12D/+}$; $Trp53^{fl/fl}$ (KP) mice (*Figure 1D*). The expansion of Msi2 expression in lung tumors compared to normal lung epithelia suggests that normal cells expressing Msi2 are more sensitive to transformation, or that once transformed, Msi2 expression is amplified in tumor cells.

### Msi2-expressing cells are more responsive to transformation by mutant Ras and p53

To determine whether Msi2 expression marks cells with increased capacity for tumor growth, we generated Msi2-reporter $Kras^{G12D/+}$; $Trp53^{fl/fl}$ mice with inducible Cre expression ($Msi2^{eGFP/+}$; $Kras^{G12D/+}$; $Trp53^{fl/fl}$; $Rosa26^{CreERT2/+}$; *Figure 2A*), delivered tamoxifen, isolated Msi2⁺ and Msi2⁻ lung epithelial cells, and tested sphere-forming capacity. As shown in *Figure 2B*, Msi2⁺ cells had enhanced sphere-forming capacity, with a sixfold increase in sphere-forming observed at primary passage, and this capacity was

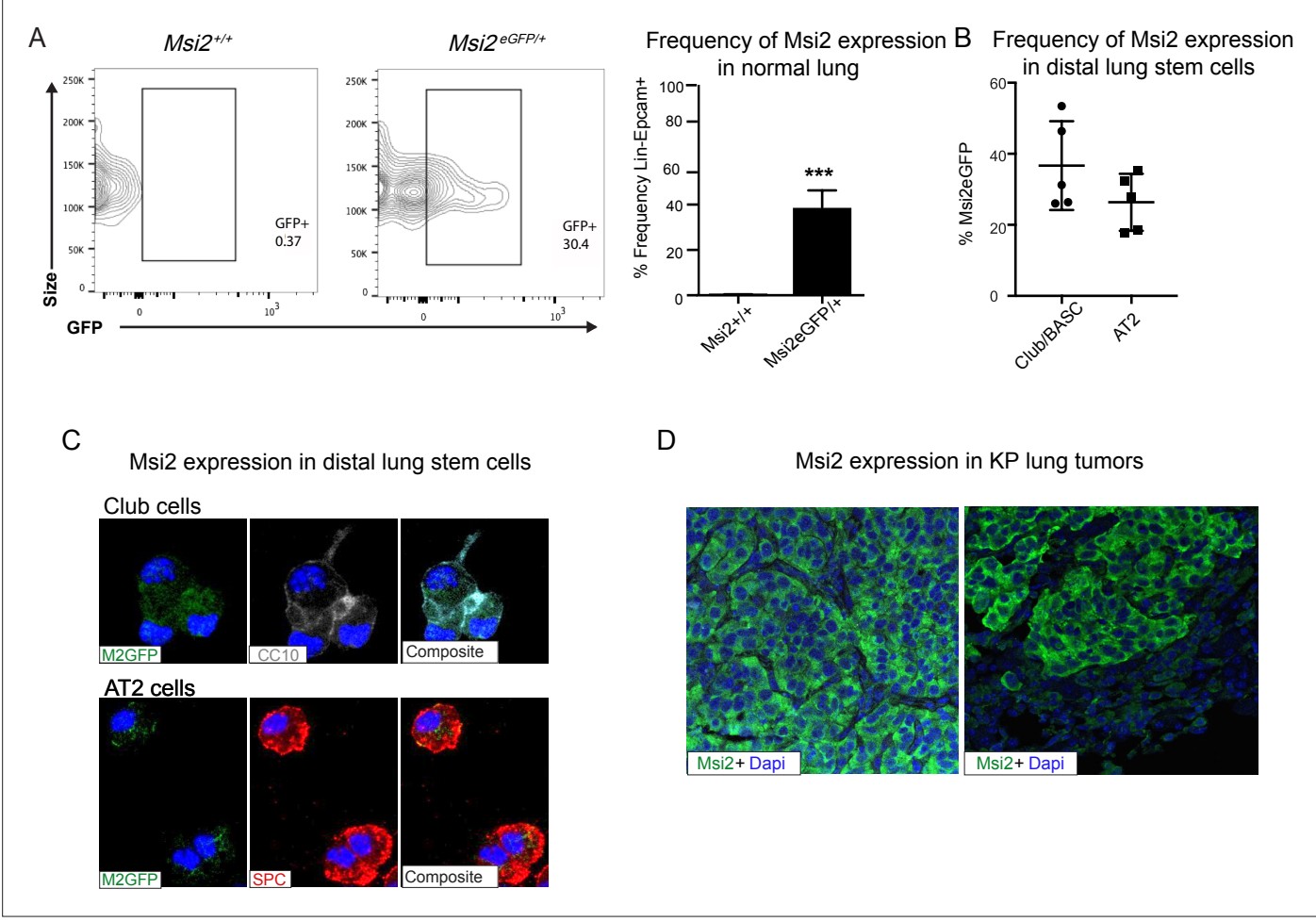

**Figure 1.** Msi2 is expressed in the stem cells of the distal lung and in lung adenocarcinoma. (**A**) Msi2 is expressed in the Lin⁻EpCAM⁺ lung epithelial cells of *Msi2^GFP/+* (M2GFP) reporter mice. Representative FACS (Fluorescence activated cell sorting) plots shown for non-reporter (*Msi2^+/+*, left) and M2GFP reporter mouse lungs (*Msi2^eGFP/+* middle). Frequency of M2GFP-expressing cells in lung epithelia (right; *n* = 2 non-reporter, *n* = 5 reporter). Data represented as mean ± SD, one outlier identified and removed using the Grubb's test, ***p < 0.001 by Student's *t*-test. (**B, C**) Msi2 is expressed in the known stem cell populations of the distal lung. (**B**) Frequency of Msi2 expression in the Lin⁻EpCAM⁺Sca1⁺ enriched Club/BASC and the Lin⁻EpCAM⁺Sca1⁻ enriched AT2 cell populations (*n* = 5). (**C**) Representative images of cytospins from M2GFP reporter mouse Lin⁻EpCAM⁺ lung epithelial cells showing expression of Msi2 expression in Club cells (top row, marked by CC10 staining) and AT2 cells (bottom row, marked by SPC staining). (**D**) Representative images of Msi2 in tumors from KP mice; some tumors display ubiquitous expression of Msi2 (left), while others display more heterogeneous expression (right).

retained for up to four passages (**Figure 2B**). Furthermore, spheres that were isolated and transplanted subcutaneously into the flanks of immunocompromised mice developed Msi2-expressing tumors in vivo as well as metastatic lesions in the lung, demonstrating that the observed sphere-forming capacity is reflective of tumorigenicity and not merely stem cell capacity (**Figure 2C–E**). To understand the basis for the enhanced tumorigenic capacity of Msi2⁺ cells, we carried out RNAseq to examine the transcriptional landscape of Msi2-expressing cells in normal lung epithelia. Interestingly, the differentially expressed gene profile showed that Msi2⁺ cells are transcriptionally distinct from Msi2⁻ cells (**Figure 2—figure supplement 1A**). Gene Set Enrichment Analysis (GSEA) further revealed that Msi2⁺ cells are significantly enriched for signatures related to developmental and stem cell signaling, consistent with the known role of Msi2 as a critical regulator of the stem cell state (**Okano et al., 2002**; **Hope et al., 2010**; **Kharas et al., 2010**), as well as signatures related to oncogenic signaling and therapy resistance (**Figure 2—figure supplement 1B–D**) which are associated with aggressive cancers. Over 1900 differentially expressed genes were found to be enriched in the Msi2⁺ population. In line with the GSEA results, these enriched genes included those involved in developmental and stem cell signaling (such as *Dll1*, *Jag2*, and *Notch3*) (**Kumar et al., 2019**; **Than-Trong et al., 2018**; **Ntziachristos et al.,**

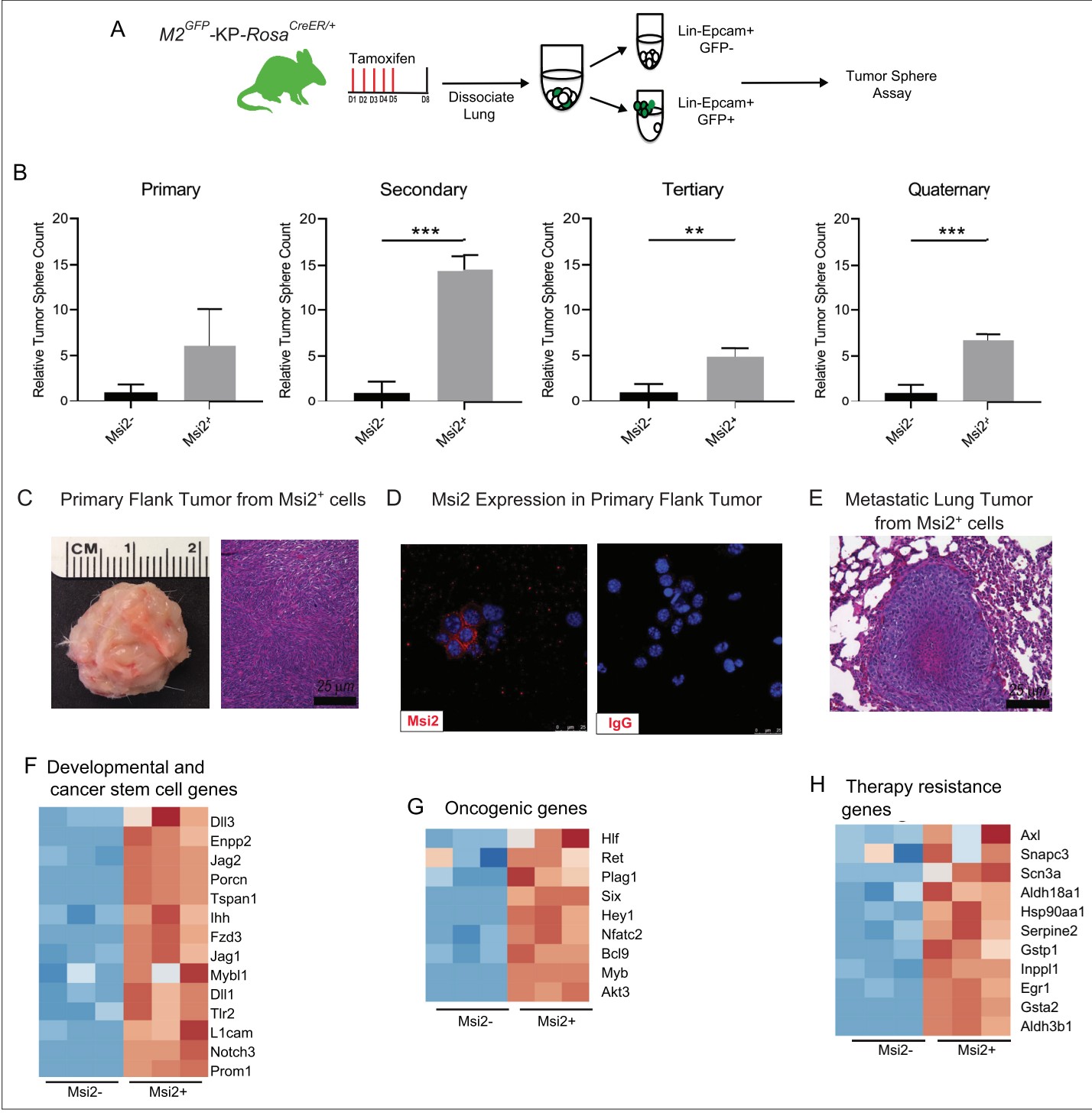

**Figure 2.** Msi2-expressing cells preferentially favor tumor growth in lung adenocarcinoma. (**A**) Schematic of the strategy used to measure the tumor sphere-forming capacity of Msi2-expressing and non-expressing cells in vitro. $Msi2^{GFP/+}$; $Kras^{G12D/+}$; $Trp53^{fl/fl}$; $Rosa26^{CreERT2/+}$ ($M2^{GFP}$-KP-$Rosa^{CreER/+}$) mice were treated with tamoxifen for 5 days to induce recombination of floxed alleles. Three days following the final dose of tamoxifen, lungs were dissociated and Msi2-expressing (GFP+) or non-expressing (GFP−) cells were isolated by FACS and then plated in a tumor sphere assay. (**B**) Msi2-expressing cells isolated from KP lung epithelia following tamoxifen treatment preferentially form tumor spheres over multiple cell passages in vitro as compared to non-expressing cells. Representative experiment shown ($n$ = 3). Data represented as mean ± SD, **p < 0.01, ***p < 0.001 by Student's $t$-test. (**C–E**) Tumor spheres formed by Msi2+ cells in vitro form aggressive tumors in vivo. Tumor spheres isolated from Msi2+ cells after quaternary passage in vitro and transplanted in vivo form highly aggressive flank tumors (**C**) that retain Msi2 expression (**D**) and are able to metastasize to the lung (**E**). (**F–H**) Gene Set Enrichment Analysis of Msi2-expressing normal lung epithelial cell gene signatures. Heatmaps of selected genes that are involved

*Figure 2 continued on next page*

Figure 2 continued

in developmental and stem cell signaling (**F**), oncogenic signaling (**G**), and therapy resistance (**H**). Red represents genes that are upregulated in the presence of Msi2, while blue represents genes that are downregulated in the absence of Msi2.

The online version of this article includes the following figure supplement(s) for figure 2:

**Figure supplement 1.** Msi2-expressing cells preferentially favor tumor growth in lung adenocarcinoma.

*2014*; *Yang et al., 2011*), oncogenic signaling (such as *Akt3*, *Ret*, *Myb*, and *Fos*) (*Chin et al., 2014*; *Mulligan, 2014*; *Ramsay and Gonda, 2008*; *Muhammad et al., 2017*), and therapy resistance (such as *Axl*, *Egr1*, and several glutathione *S* transferases implicated in platinum drug resistance including *Gstp1 and Gsta2*) (*Pinato et al., 2019*; *Allocati et al., 2018*; *Kumar et al., 2017*; *Figure 2F–H*). These findings suggest that prior to tumor initiation, the transcriptional landscape of Msi2-expressing cells is enriched in signaling programs that are conducive to tumorigenesis, which may make them uniquely poised for transformation.

## Genetic deletion of Msi2 leads to a decrease in tumor incidence, burden, and progression

To determine if Msi2 not only marks a cell with an enhanced capacity for lung adenocarcinoma growth, but whether it may also be required for initiation of these tumors, we compared wild-type and Msi2-knockout $Kras^{G12D/+}$; $Trp53^{fl/fl}$ mice ($Msi2^{-/-}$; $Kras^{G12D/+}$; $Trp53^{fl/fl}$) (*Figure 3A*). In this model, the *Msi2* gene has been disrupted via gene trap mutagenesis (*Ito et al., 2010*), while the inhalation of adeno-viral-Cre (Ad-Cre) results in the conditional activation of Kras and inactivation of p53 in the lung (*DuPage et al., 2009*). Importantly, loss of Msi2 led to the formation of significantly fewer (84%) tumor lesions as well as a significantly lower (85%) overall tumor burden in the lung at a 14-week midpoint (*Figure 3B–D*). Furthermore, while there was no difference in the frequency of low-grade tumors, there was a significant decrease in the frequency of mid- and high-grade tumors (45–54%), suggesting that loss of Msi2 delayed tumor progression from low to high grade (*Figure 3E*). Importantly, $Msi2^{-/-}$ KP mice infected with Ad-Cre developed tumors less frequently than their wild-type counterparts (61% $Msi2^{-/-}$ vs 83% $Msi2^{+/+}$; *Figure 3F*). To confirm that the population of lung stem cells is not reduced in Msi2-knockout mice and therefore contributing to a decrease in tumor lesions, we stained the lungs of WT and Msi2-knockout mice for Club/BASC and AT2 lung stem cells and saw no difference (data not shown). Additionally, $Msi2^{-/-}$ KP mice displayed a significant increase in overall survival compared to $Msi2^{+/+}$ KP mice. $Msi2^{+/+}$ KP mice had a median survival of 209 days while the $Msi2^{-/-}$ KP mice had a median survival of 428 days, reflecting a twofold increase in survival (p = 0.04, *Figure 3G*). Notably, the majority of lung tumors that formed in $Msi2^{-/-}$ KP mice were found to express Msi2, suggesting that the tumors that formed in $Msi2^{-/-}$ KP mice were escapers that re-expressed Msi2, underscoring the dependency on Msi2 signaling for tumor formation (*Figure 3H*). Taken together these results suggest that Msi2 not only marks cells with an enhanced capacity for tumor growth but is also required for initiation and progression of lung adenocarcinoma.

The analyses described above were performed in $Msi2^{-/-}$ KP mice, and therefore in an environment in which Msi2 is absent at the onset of tumorigenesis. To test whether established cancers have a similar dependency on Msi2 for their growth, we knocked down Msi2 in primary tumor cell lines generated from $Kras^{G12D/+}$; $Trp53^{fl/fl}$ lung tumors (KP cell line) and measured tumorsphere formation. Importantly, loss of Msi2 led to a significant decrease (75%) in tumorsphere formation in vitro, suggesting that Msi2 is required for tumor cell propagation (*Figure 4A, B*). To confirm this finding in vivo, Msi2-knockdown KP cells were transplanted into the flanks of syngeneic mice, and tumor growth monitored over time. As shown in *Figure 4*, the Msi2-knockdown tumors had significantly reduced (89%) tumor cell numbers (*Figure 4C, D*), indicating that established tumors remain dependent on Msi2 signaling for growth.

To determine the relevance of our findings for human disease, we examined the effect of Msi2 inhibition on primary patient samples. To this end, patient samples were transplanted into immuno-compromised mice to generate PDXs. Once established, PDX tumors were harvested, dissociated, transduced with either shControl or shMsi2 lentivirus, and subsequently transplanted into the flanks of immunocompromised mice (*Figure 4E*). Importantly, although an equivalent number of shControl and shMsi2 cells were transplanted into recipient mice, there was a notable delay in the growth of

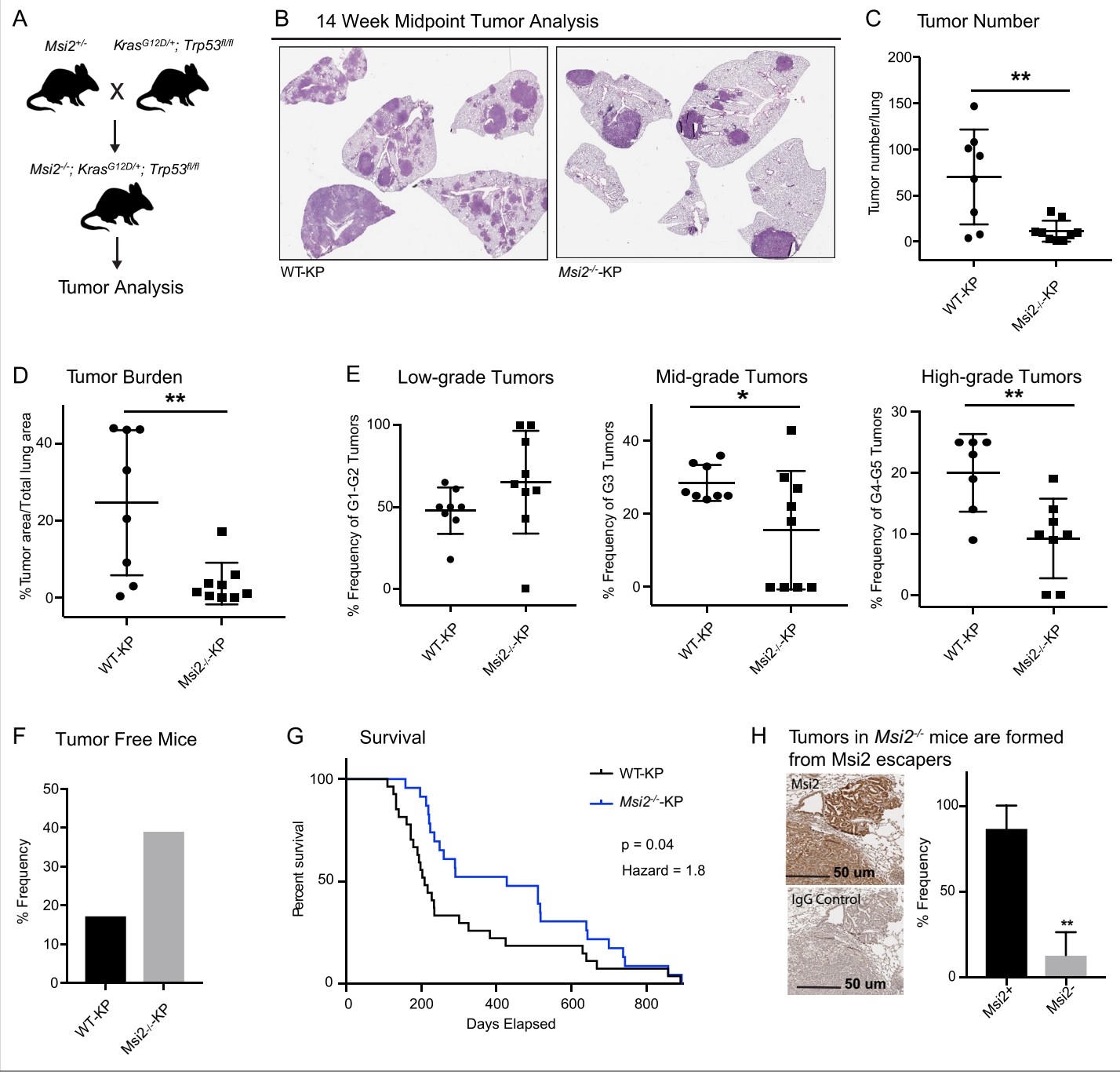

**Figure 3.** Msi2 is required for lung adenocarcinoma growth and progression. (**A**) Breeding scheme used to generate the $Msi2^{-/-}$; $Kras^{G12D/+}$; $Trp53^{fl/fl}$ ($Msi2^{-/-}$-KP) model of lung adenocarcinoma. (**B**) Representative images of wild-type (left) and $Msi2^{-/-}$ (right) $Kras^{G12D/+}$; $Trp53^{fl/fl}$ lungs gender and age matched mice at 14 weeks after tumor initiation. The number of tumors formed (**C**) and overall tumor burden (**D**) is significantly reduced in $Msi2^{-/-}$ KP mice. (**E**) $Msi2^{-/-}$ KP mice have a significant reduction in the frequency of mid- and high-grade tumors. Data represented as mean ± SD, two outliers identified and removed for high-grade tumors using the Grubb's test *p < 0.05, **p < 0.01 by Student's t-test. (**F**) The tumor take rate is reduced in $Msi2^{-/-}$ KP mice. While only 17% of AdCre-infected wild-type mice are tumor-free at the time of death, more than twice as many (39%) $Msi2^{-/-}$ KP mice are tumor-free at the time of death. (**G**) $Msi2^{-/-}$ KP have significantly increased survival (p = 0.04, hazard ratio = 1.8; n = 27 WT-KP, n = 23 $Msi2^{-/-}$-KP), with a median survival of 428 days compared to 209 days for wild-type mice. Log-rank test was used to determine the difference in survival curves between wild-type and $Msi2^{-/-}$ KP mice. (**H**) Lung tumors in $Msi2^{-/-}$ KP mice express Msi2. Representative immunohistochemical staining (left) for Msi2 (top) or IgG control (bottom) in lung tumors from a 16-week-old $Msi2^{-/-}$ KP mouse. Quantification of the frequency of Msi2-expressing tumors from $Msi2^{-/-}$ KP mice shows that while some tumors lack Msi2 expression the majority of tumors express Msi2 (right, n = 3). Data represented as mean ± SD, **p < 0.01 by Student's t-test.

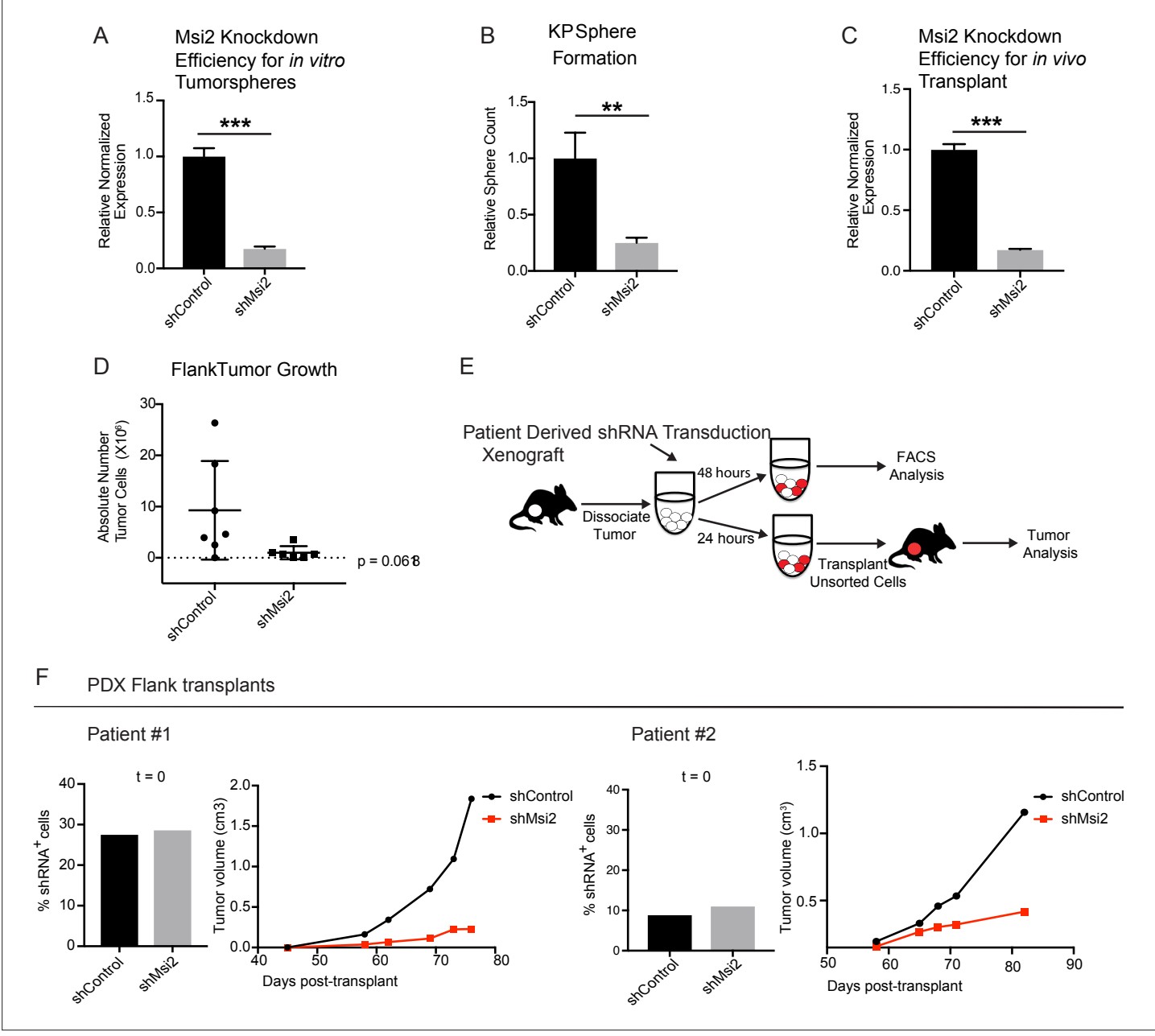

**Figure 4.** Loss of Msi2 impairs growth of established cancer. (**A**) Mouse KP cells transduced with shMsi2 and used for in vitro tumorsphere assays have an 83% reduction in Msi2 mRNA expression. (**B**) Loss of Msi2 significantly impairs tumor sphere formation in vitro ($n = 3$). (**C**) Mouse KP cells transduced with shMsi2 and used for in vivo flank transplant assays have an 83% reduction in Msi2 mRNA expression. (**D**) Loss of Msi2 consistently impairs tumor growth in flank transplants in vivo. Data represented as mean ± SD, one outlier identified and removed using the Grubb's test **p < 0.01 by Student's t-test. (**E**) Schematic for testing the impact of Msi2 loss on growth of patient-derived xenografts in vivo. Patient-derived xenografts were harvested, dissociated, transduced with shMsi2 or shControl virus and split into two pools that were incubated for either 24 or 48 hr. After 24 hr, one pool of cells (containing a mixture of transduced and untransduced cells) was transplanted into the flanks of NSG mice. After 48 hr the remaining pool of cells was analyzed via FACS to determine the frequency of cells infected with shRNA. (**F**) The frequency of infection was comparable for shControl and shMsi2 in two independent patient samples. Reduced *Msi2* expression led to pronounced inhibition of tumor growth in two independent patient-derived xenografts.

The online version of this article includes the following figure supplement(s) for figure 4:

**Figure supplement 1.** FACS plots for patient-derived xenografts (PDXs) transduced with shRNA PDX cells transduced with either shControl or shMsi2 lentivirus have a similar frequency of infection.

Msi2-knockdown tumors, which were 64–88% smaller than control tumors at endpoint (*Figure 4F*, *Figure 4—figure supplement 1*). These findings suggest that human lung adenocarcinomas are dependent on Msi2 for their continued growth.

## Msi2 regulates signaling pathways critical for tumor growth

To understand the mechanism by which Msi2 impacts tumor growth, we knocked down Msi2 in KP cell lines and carried out an RNAseq analysis comparing control and knock-down cells. Principal component analysis shows a clear separation between the control and Msi2-knockdown cells and significant changes in over ~170 genes, highlighting the marked impact that loss of Msi2 has on the transcriptional profile of lung adenocarcinoma (*Figure 5A*). Consistent with the role of Msi2 in stem cell maintenance, the inhibition of Msi2 in lung cancer cells had a significant impact on developmental and stem cell signaling programs and genes, this included *Porcn*, the transcription regulator *Nupr1*, and the NuRD complex member *Mbd3*, which have been shown to play a role in the maintenance of the stem cell state (*Figure 5B, C*; *Clevers et al., 2014*; *Tammela et al., 2017*; *Zhou et al., 2018*; *Loughran et al., 2017*). In addition, Msi2 impacted a broad range of other signaling programs and their associated genes, including DNA repair and metabolism. DNA repair genes included *Brca1*, *Atm*, and several Fanconi anemia complementation group genes (*Scully et al., 1997*; *Bakkenist and Kastan, 2003*; *Walden and Deans, 2014*), and metabolism programs included the gluconeogenesis enzyme *Pck2*, which has been shown to promote lung cancer cell survival in low glucose conditions, several isoforms of *Cpt1*, a key regulator in fatty acid oxidation known to promote tumor growth in a variety of cancers, and the glutamate dehydrogenase *Glud1*, which has been shown to promote tumorigenesis in lung cancer xenografts (*Figure 5D–G*; *Leithner et al., 2015*; *Kuo and Ann, 2018*; *Jin et al., 2015*). Interestingly, several known regulators of lung adenocarcinoma such as *Gli1*, a canonical downstream effector of the Hedgehog signaling pathway, βIII tubulin (*Tubb3*), and CC Chemokine Receptor 1 (*Ccr1*) were found to be significantly downregulated following Msi2 inhibition as well (*Figure 5H*; *Rentas et al., 2016*; *Song et al., 2011*; *Han et al., 2017*; *Po et al., 2017*; *Avis et al., 1996*; *Wang et al., 2009a*; *Wang et al., 2009b*; *Stallings-Mann et al., 2012*; *McCarroll et al., 2015*). These data suggest that Msi2 is a regulator of a variety of signaling pathways that are critical to tumor growth and lung cancer progression. In addition to these known regulators, a number of genes not previously implicated in lung cancer were found to be significantly downregulated following Msi2 inhibition, suggesting that they may play a role in tumor growth (*Figure 5I, J*). To identify potential novel regulators of lung adenocarcinoma we focused on significantly downregulated genes that were highly expressed, which narrowed our gene list to 30 potential candidates. PCR-based validation and prioritization of those genes with no known role in lung cancer led to a subset that included genes such as the E3 ubiquitin ligase *Rnf157*, which promotes neuronal growth through the inhibition of apoptosis (*Matz et al., 2015*), Synaptotagmin 11 (*Syt11*), a critical mediator of neuronal vesicular trafficking (*Shimojo et al., 2019*), prostoglandin D2 synthase (*Ptgds*) known to be involved in reproductive organ development (*Malki et al., 2005*), and ADP ribosylation like binding protein (*Arl2bp*), a known regulator of STAT3 signaling (*Muromoto et al., 2008*). To test whether these genes regulate tumor cell growth we knocked down each of these genes (*Ptgds*, *Arl2bp*, *Rnf157*, and *Syt11*) in the KP cell line. Inhibition of each target led to a significant decrease (88–98%) in the formation of tumor spheres in vitro (*Figure 5K*). These findings not only identify these genes as downstream effectors of Msi2 activity but more generally show that this dataset could be an important resource to identify new regulators of lung adenocarcinoma growth.

## Discussion

The stem cell fate determinant Msi2 is commonly found to be expressed in carcinomas (*Moreira et al., 2010*; *Kharas and Lengner, 2017*; *Uhlén et al., 2015*; *Uhlen et al., 2010*) and Msi signaling has been shown to contribute to colon adenomas (*Wang et al., 2015*) and maintain the aggressive, undifferentiated state in hematologic malignancies and pancreatic cancer (*Ito et al., 2010*; *Fox et al., 2016*). Here, we have used genetically engineered and autochthonous models to show that Msi2 is critically required in the development and progression of primary lung adenocarcinoma. Msi2 signaling emerges as a key dependency at initiation and continues to be needed during tumor progression. Using the well-established *Kras*$^{G12D}$; *Trp53*$^{fl/fl}$ (KP) mouse model of lung adenocarcinoma (*Jackson*

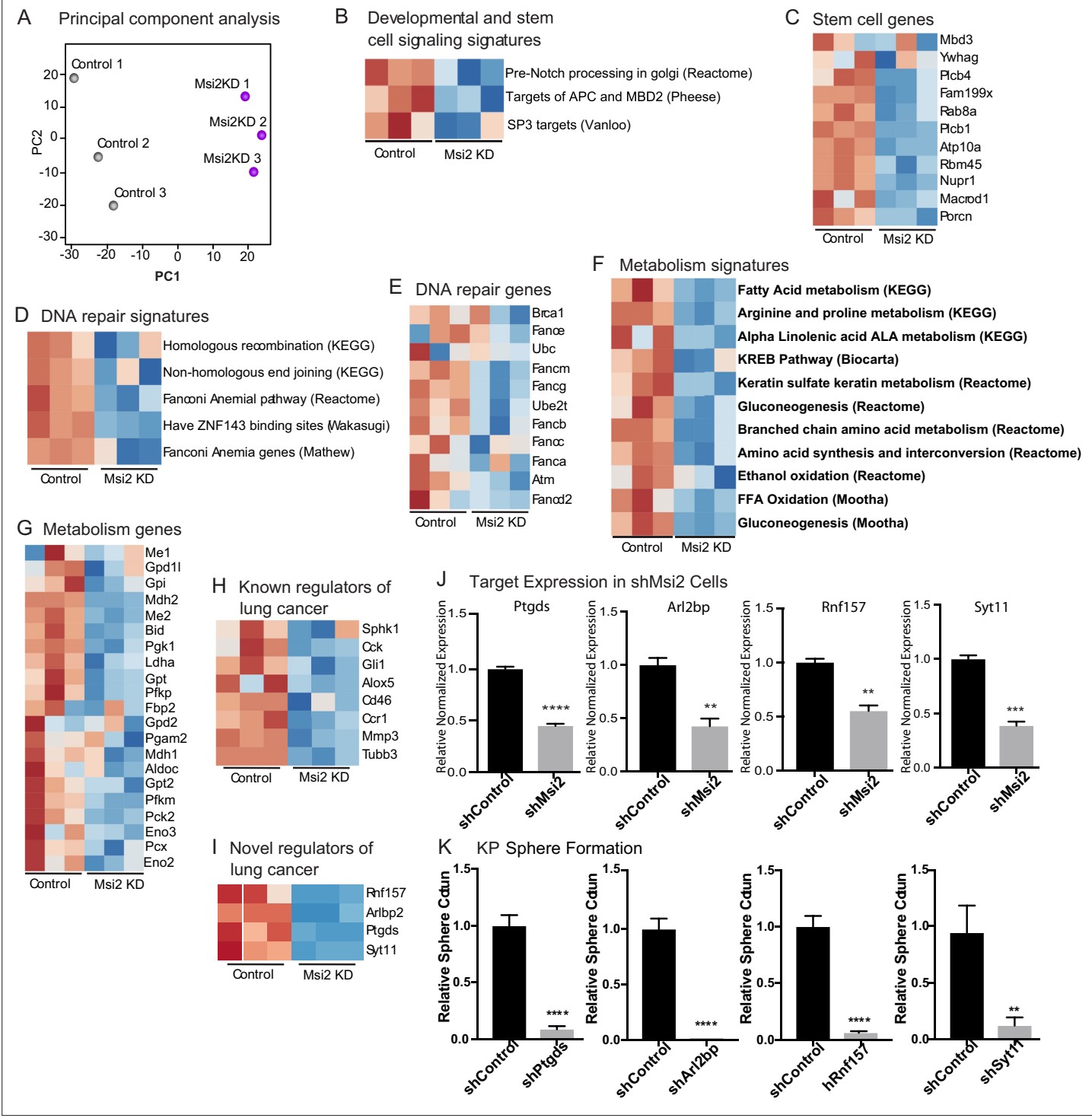

**Figure 5.** Msi2 regulates oncogenic signaling in lung adenocarcinoma. (**A**) Principal component analysis of KP cells transduced with shMsi2 (Msi2KD, purple) or shControl (gray). (**B–G**) Gene Set Enrichment Analysis of Msi2KD gene signatures. Heatmaps of gene signatures and selected genes that are involved in developmental and stem cell signaling (**B, C**), DNA repair (**D, E**), and metabolism (**F, G**) and are downregulated (shown in blue) following the loss of Msi2. Known (**H**) and novel (**I**) regulators of lung cancer that are downregulated following loss of Msi2. (**B–I**) Red represents genes or gene signatures that are upregulated in the presence of Msi2, while blue represents genes that are downregulated following the loss of Msi2. (**J**) Confirmation of downregulated genes in Msi2KD cells using qRT-PCR analysis. (**K**) In vitro functional analysis of novel effectors of lung cancer (Ptgds, Arl2bp, Rnf157, and Sty11) downregulated by Msi2. KP cells were transduced with shRNA to inhibit the genes of interest and analyzed for the resulting impact on tumor sphere formation in vitro. Sphere formation, n = 3 per condition. Data represented as mean ± SD, **p < 0.01, ***p < 0.001, ****p < 0.0001 by Student's t-test.

*et al., 2005*), we created a Msi2-knockout lung adenocarcinoma model and found that deletion of Msi2 at initiation significantly impaired tumor incidence, reduced tumor burden and progression, and cumulatively led to a doubling of median survival. Furthermore, loss of Msi2 impaired propagation of established KP tumors in vitro and in vivo. In order to determine whether these findings were relevant for the human disease, we used PDXs, which have proven to be a highly valuable tool for understanding the biology of human cancers and have been shown to faithfully predict therapeutic response in a majority of patients (*Tentler et al., 2012*; *Siolas and Hannon, 2013*; *Hidalgo et al., 2014*). Inhibition of Msi2 in PDXs also led to impaired tumor growth in vivo, suggesting that dependence on Msi2 is conserved in human lung adenocarcinoma.

Our finding that the loss of Msi2 prior to transformation significantly impacts tumor initiation prompts the interesting question of whether Msi2 signaling can influence early tumor growth for lung adenocarcinoma. Previous use of the KP model has illustrated how both the cellular and molecular context at the time of transformation influences subsequent cancer development (*Sutherland et al., 2014*). Using cell-type-specific Ad-Cre to drive the expression of either *Kras^{G12D}* alone or *Kras^{G12D}* in combination with *Trp53^{fl/fl}*, work by *Sutherland et al., 2014* has shown that the cell of origin for adenocarcinoma differs based on the genetic mutations used to drive the cancer. When *Kras^{G12D}* alone was used as the driver, AT2 cells were able to form adenomas which then progressed to adenocarcinomas, while Club cells were only able to form adenomas which did not progress further. However, when *Kras^{G12D}* was used in combination with *Trp53^{fl/fl}*, both cell types were capable of forming adenomas which could then progress to adenocarcinomas. While both genetic drivers and the cell of origin are key components for tumor formation, whether there are signaling factors that endow the cell of origin with the capacity to transform when presented with these key drivers is an area that has been less explored. In this context, our work sheds new light on such signals required at initiation as we found that Msi2 is expressed in a subpopulation of distal lung stem cells, that cells expressing Msi2 are preferentially sensitive to transformation, and that its genetic ablation leads to a marked decrease in tumor initiation and progression. The results of our RNAseq analysis suggest that this enhanced capacity may be established through an enrichment in signaling programs involved in the maintenance of an undifferentiated state as well as those known to support aggressive tumor growth which then provides a permissive environment for early tumor growth and development. These findings identify a new role for Msi2 signaling as an enhancer of tumor growth for lung adenocarcinoma and indicate that subpopulations harboring these signals may be particularly primed for transformation.

The continued dependency on Msi2 signaling following tumor initiation illustrates the potency of Msi2 as a regulator of tumor progression; thus, defining the mechanism by which Msi2 exerts this influence in lung adenocarcinoma would be critical to understanding the molecular basis of its broad influence. Here, our RNAseq analysis suggests that Msi2 may act in part as a regulator of the stem cell state. Interestingly, this analysis also showed that the role of Msi2 may extend to the regulation of processes crucial for tumor growth, such as metabolism and DNA repair, thus placing it upstream of multiple potent oncogenic signals. This work also led to the identification of several novel effectors of lung adenocarcinoma regulated by Msi2. Our functional experiments inhibiting these genes showed that loss of *Ptgds*, *Arl2bp*, *Rnf157*, and *Syt11* can block lung cancer sphere formation. Previous work has shown that Ptgds indirectly facilitates phosphorylated Sox9 nuclear translocation via activation of cAMP-dependent protein kinase A (PKA) (*Malki et al., 2005*) and is important in the development of Sertoli cells. While *Ptgds* has been shown previously to be expressed in human lung tumors (*Ragolia et al., 2010*) our work is the first to identify a functional role of Ptgds in lung adenocarcinoma. As a regulator of STAT3 signaling, Arl2bp directly binds to the STAT3 transcription factor, which maintains STAT3's phosphorylation status and its localization in the nucleus (*Muromoto et al., 2008*). STAT3 activity has been implicated in NSCLC (*Mohrherr et al., 2020*), thus modulating STAT3 activity through *Arl2bp* inhibition may provide an alternative therapeutic strategy. Lastly, both Rnf157 and Syt11 are predominantly expressed in the brain and have been implicated in neuronal cell survival and dendrite growth and neuronal vesicular trafficking, respectively (*Matz et al., 2015*; *Shimojo et al., 2019*). Thus far, their roles have not been investigated in cancer. Our findings suggest the need for further study of these genes and raise the possibility that they could serve as future therapeutic targets.

While early treatment approaches centered principally on the use of cytotoxic agents, the treatment of lung cancer has evolved to incorporate the use of targeted therapies and immunotherapy, when appropriate, in combination with the standard cytotoxic regimen (*Herbst et al., 2018*). For

patients with a lung cancer subtype harboring known targetable mutations, this has led to considerable improvements in the response to treatment. Yet troubling survival rates, driven largely by either late-stage detection or recurrence of the disease—even in cases for which a targeted therapy has been offered— illustrate the need for an increased understanding of the biology of lung cancer in order to inform future therapeutic approaches. The findings presented here identify a novel role for Msi2 in the growth and progression of lung adenocarcinoma and suggest that this role extends to human disease. Furthermore, these findings identify novel effectors of lung adenocarcinoma regulated by Msi2 thereby lending new insight into the mechanism by which Msi2 exerts its influence on this disease. These new insights enhance our understanding of the biology of lung adenocarcinoma and may serve to open new avenues of focus for future therapeutic strategies.

## Methods

### Mice

*Msi2eGFP/+* knock-in reporter mice were generated as previously described (*Fox et al., 2016*) and are heterozygous for the *Msi2* allele. The Msi2 mutant mouse, B6; CB-Msi2Gt(pU-21T)2Imeg (Msi2−/−) was established by gene trap mutagenesis as previously described (*Ito et al., 2010*). The LSL-Kras G12D mouse, B6.129S4-Krastm4Tyj/J (stock number 008179), the *Trp53flox* mouse, B6.129P2-Trp53tm1Brn/J (stock number 008462), and the R26-CreERT2 mouse, B6;*129-Gt(ROSA) 26Sortm1(cre/Esr1)Tyj* (stock number 008463) were purchased from The Jackson Laboratory. At 9–16 weeks of age $5 \times 10^6$ PFU Adeno-Cre (purchased from the University of Iowa) was delivered by intratracheal instillation as previously described (*DuPage et al., 2009*). Where indicated, adult mice were administered tamoxifen (Sigma) in corn oil (20 mg/ml) daily by intraperitoneal injection (150 mg/kg of body weight) for 5 consecutive days. Lungs were then isolated 3 days following the last dose of tamoxifen. Mice were bred and maintained in the animal care facilities at the University of California San Diego. All animal experiments were performed according to protocols approved by the University of California San Diego Institutional Animal Care and Use Committee.

### Tissue dissociation and FACS analysis

For isolation of non-tumorigenic lung tissue, tissue dissociation was performed as described (*Kim et al., 2005*). Briefly, mice were anesthetized, perfused with cold phosphate-buffered saline (PBS) using a peristaltic pump (Fisher Scientific), and 2 ml dispase (Corning) was added to the lungs via intratracheal instillation. The lungs were then removed, placed on ice, and the lobes were minced. For isolation of lung tumors, mice were anesthetized and perfused with PBS as described, then lungs were removed and placed on ice. Visible tumors were dissected away from surrounding tissue, placed on ice in 2 ml dispase, and minced. Minced non-tumorigenic lung tissue or grossly dissected lung tumors were then added to 2 mg/ml collagenase/dispase in PBS (Millipore Sigma), placed in a rotisserie, and rotated for 45 min at 37°C. 25 ng/ml DNAseI (Sigma-Aldrich) was added, and the tissue was then passed through a 100- and 40-µm filter (Corning), and centrifuged at 1200 rpm for 5 min at 4°C. Cells were then treated with 1 ml red blood cell lysis buffer (Thermo Fisher) on ice for 90 s, washed, and resuspended in HBSS (Gibco, Life Technologies) containing 5% fetal bovine serum (FBS) and 2 mM EDTA before staining. Analysis and cell sorting was performed using a FACSAriaIII machine (Becton Dickinson) and data were analyzed using FlowJo software (Tree Star). The following antibodies (1 µl/1 M cells) were used: rat anti-mouse EpCAM-APC (eBioscience, #17-5791-82, clone G8.8), CD45-PE (eBioscience, #12-0454-82, clone 104), CD45-PeCy7 (eBioscience, #25-0451-82, clone 30-F11), CD31-PE (eBioscience, #12-0311-83, clone 390), CD31-PeCy7 (eBioscience; #25-0311-81, clone 390), and Sca1-PeCy7 (eBioscience, #25-5981-82, clone D7). CD31 and CD45 antibodies were used to mark the Lineage+ populations of the lung.

### Histology, immunohistochemistry, and immunofluorescence

For the isolation of tumor-bearing lungs from KP mice, mice were anesthetized, perfused with PBS as described, and 4% paraformaldehyde (PFA; Fisher Scientific) was added to the lungs via intratracheal instillation. Lungs were then removed, placed in 4% PFA overnight at 4°C, and stored in 70% ethanol prior to processing. The fixed lung tissue was then paraffin embedded at the UCSD Histology and Immunohistochemistry Core at The Sanford Consortium for Regenerative Medicine

according to standard protocols, and 5 µm sections were obtained. Paraffin embedded tissues were then deparaffinized in xylene and rehydrated in a 100%, 95%, 70%, 50%, and 30% ethanol series. For immunohistochemistry, endogenous peroxidase was quenched in 3% hydrogen peroxide (Fisher Scientific) for 30 min at room temperature prior to antigen retrieval. Antigen retrieval was performed for 30 min in 95–100 °C 1× citrate buffer, pH 6.0 (eBioscience). Sections were blocked in TBS (Tris buffered saline) or PBS containing 0.1% Triton X-100 (Sigma-Aldrich), 10% goat or donkey serum (Sigma-Aldrich), and 5% bovine serum albumin. For single-cell suspensions of lung epithelia or lung tumors, Lin⁻EpCAM⁺ cells were isolated via FACS in the manner described previously. Cells were resuspended in DMEM:F12 (Fisher Scientific) supplemented with 50% FBS and adhered to slides by centrifugation at 500 rpm. Twenty-four hours later, cells were fixed in 4% PFA (Fisher Scientific), washed in PBS, and blocked with PBS containing 0.1% Triton X-100 (Sigma-Aldrich), 10% Goat serum (Fisher Scientific), and 5% bovine serum albumin (Invitrogen).

All incubations with primary antibodies were carried out overnight at 4°C. For immunofluorescence staining, incubation with Alexafluor-conjugated secondary antibodies (Molecular Probes) was performed for 1 hr at room temperature. DAPI (Molecular Probes) was used to detect DNA and images were obtained with a Confocal Leica TCS SP5 II (Leica Microsystems). For immunohistochemical staining, incubation with biotinylated secondary antibodies (Vector Laboratories) was performed for 45 min at 20–25°C. Vectastain ABC HRP Kit (Vector Laboratories) was used according to the manufacturer's protocol. Sections were counterstained with hematoxylin. The following primary antibodies were used: chicken anti-GFP (Abcam, ab13970; 1:200), rabbit anti-Msi2 (Abcam, ab76148; clone EP1305Y, 1:500), rabbit anti-SPC (Santa Cruz, sc-13979; clone FL-197, 1:100), and goat anti-CC10 (Santa Cruz, sc-9773; clone S-20, 1:200). Whole slide imaging was used to determine the number of tumors, tumor burden, and tumor grade in the tumor-bearing lungs of KP mice. H&E-stained slides were scanned using Aperio AT2 digital whole slide scanning (Leica Biosystems), and tumor analysis was performed using ImageScope (Aperio). Tumor grade was scored according to the criteria described by *Jackson et al., 2005*.

## Mouse lung cancer cell lines

Mouse primary lung cancer cell lines were established from end-stage KP mice as follows: tumors were isolated and dissociated into single-cell suspensions in the manner described above, then plated in 1× DMEM:F12 (Fisher Scientific) supplemented with 10% FBS, 1× pen/strep (Fisher Scientific), 1× N-2 supplement (Life Technologies), 35 µg/ml bovine pituitary extract (Fisher Scientific), 20 ng/ml recombinant mouse EGF (BioLegend), 20 ng/ml recombinant mouse FGF (R&D Systems). Upon reaching 80% confluency the cells were collected and resuspended in HBSS (Gibco, Life Technologies) containing 5% FBS and 2 mM EDTA, treated with FC block, and stained with EpCAM-APC (eBioscience), and CD31-PE (eBioscience). CD31−EpCAM+ cells were sorted and plated for at least one additional passage prior to use in functional assays. Functional assays were performed using cell lines between passages 2 and 6.

## Patient-derived xenografts

Patient pleural effusate samples were obtained from Moores Cancer Center at the University of California San Diego from Institutional Review Board-approved protocols with written informed consent in accordance with the Declaration of Helsinki. Effusates were centrifuged at 1200 rpm for 5 min at 4°C to pellet circulating tumor cells. Cells were then treated with 1 ml red blood cell lysis buffer (Thermo Fisher) on ice for 90 s, washed in HBSS (Gibco, Life Technologies) containing 5% FBS and 2 mM EDTA, and resuspended in 1× DMEM:F12 (Fisher Scientific) supplemented with 10% FBS, 1× pen/strep (Fisher Scientific), 1× N-2 supplement (Life Technologies), 35 µg/ml bovine pituitary extract (Fisher Scientific), 20 ng/ml recombinant human EGF (Life Technologies), and 20 ng/ml recombinant human FGF-basi (Fisher Scientific). For each patient sample, a 1:1 ratio of resuspended cells in growth factor reduced Matrigel (Corning) at a final volume of 100 µl was transplanted into the flank NSG recipient mice to allow for the development of the PDX. Tumor growth was monitored via caliper measurement, and mice were euthanized when tumors reached 2 cm in diameter. Tumors were then dissociated and used for functional analysis or for subsequent passage. Functional analysis was performed using PDXs between passages 1 and 2.

## Lung tumorsphere assays

For tumorsphere assays of mouse lung cancer cell lines involving genetic inhibition via shRNA, low passage KP cell lines were infected with shRNA lentivirus. Seventy-two hours after transduction, the cells were sorted for positively infected cells. For tumorsphere assays involving $Msi2^{eGFP/+}$; $Kras^{G12D/+}$; $Trp53^{fl/fl}$; $Rosa26^{CreERT2/+}$ mice, tamoxifen treatment and cell sorting for CD45−CD31−EpCAM+GFP+ (Msi2$^+$) and CD45−CD31−EpCAM+GFP− (Msi2$^−$) cell populations were performed as described. After sorting, cells were resuspended in full media [1× DMEM:F12 (Fisher Scientific) supplemented with 10% FBS, 1× pen/strep (Fisher Scientific), 1× N-2 supplement (Life Technologies), 35 µg/ml bovine pituitary extract (Fisher Scientific), 20 ng/ml recombinant mouse EGF (BioLegend), 20 ng/ml recombinant mouse FGF (R&D Systems)]. Resuspended cells were mixed with an equal volume of growth factor reduced Matrigel (Corning) for a total volume of 100 µl, plated in 96-well flat-bottom plates (Fisher Scientific) at a density of 500 cells/well. Matrigel was allowed to solidify at 37°C for 20 min, and 150 µl of full media was added to the well. Tumorsphere cultures were incubated at 37°C for 10–14 days with the media refreshed once a week. To passage cells, tumorspheres were dissociated from surrounding Matrigel by removing the media, adding 200 µl of 2 mg/ml collagenase/dispase (Millipore Sigma) to the wells, and incubating for 1 hr at 37°C. Cells were collected and placed on ice, the wells were washed 3× with full media, and each media collection was pooled on ice with the original cell collection. Cells were pelleted via centrifugation at 500 rcf for 5 min at 4°C, media was aspirated, 200 µl of cold Accumax (Innovative Cell Technologies) was added to the cell pellet, and cells were incubated at 37°C for 10 min. Cells were manually pipetted 100× to dissociate tumorspheres and examined via hemocytometer to determine if a single-cell suspension had been obtained. The incubation and mechanical dissociation were repeated 2–3× until a single-cell suspension was obtained. Cells were resuspended in full media, and 500 cells were plated at a 1:1 ratio with growth factor reduced Matrigel (Corning) at a final volume of 100 µl. Matrigel was allowed to solidify at 37°C for 20 min, and 150 µl of full media was added to the well, and cells were incubated in the manner described.

## Flank transplant assays

For flank transplant assays of mouse lung cancer cell lines involving genetic inhibition via shRNA, low passage KP cell lines were transduced with shRNA lentivirus, sorted, and 20,000 cells were resuspended in a 1:1 mix of full media and Matrigel (Corning) at a final volume of 100 µl in the manner described. Cells were injected subcutaneously into the flank of non-tumor bearing (i.e. not treated with Adeno-Cre), immunocompetent KP mice. Subcutaneous tumors were measured with calipers once weekly for 6–15 weeks and did not exceed 2 cm in diameter. At endpoint, flank tumors were removed, and tumors were dissociated as described. Tumor cells were stained with CD45-PeCy7 and CD31-PeCy7, and the absolute number of tumor cells was calculated by multiplying the percentage of CD45−CD31− transduced cells by the total number of CD45−CD31− live cells. For flank transplant assays of PDXs, freshly dissociated xenografts were plated in full media [1× DMEM:F12 (Fisher Scientific) supplemented with 10% FBS, 1× pen/strep (Fisher Scientific), 1× N-2 supplement (Life Technologies), 35 µg/ml bovine pituitary extract (Fisher Scientific), 20 ng/ml recombinant human EGF (Life Technologies), 20 ng/ml recombinant human FGF-basic (Fisher Scientific)]. 500,000 cells/well were plated in a 6-well plate (Fisher Scientific) coated with Matrigel (Corning) for subsequent transplant, and 83,000 cells/well were plated in a 24-well plate (Fisher Scientific) coated with Matrigel (Corning) for subsequent determination of transduction efficiency via FACS. Cells were transduced with proportionate amounts of shControl or shMsi2 lentivirus. Twenty-four hours after transduction, media from the cells intended for transplant was collected and placed on ice. 1 ml of 2 mg/ml collagenase/dispase (Millipore Sigma) was then added to the well and incubated for 45 min at 37°C to dissociate the remaining cells from Matrigel. The wells were washed 3× with full media and each wash was collected and pooled on ice with the media previously collected. Cells were pelleted by centrifugation at 1200 rpm for 5 min at 4°C, an equivalent number of shControl and shMsi2 transduced cells were resuspended in full media, mixed at a 1:1 ratio with growth factor reduced Matrigel (Corning) at a final volume of 100 µl, and transplanted subcutaneously into the flanks of NSG recipient mice. Tumor growth was monitored 1–2× a week via caliper measurement for 11–12 weeks and did not exceed 2 cm in diameter. Forty-eight hours after transduction, cells from the 24-well plate were collected in the manner described using a proportionate amount of reagents, and the frequency of transduced live cells was determined via FACS. For flank transplant

assays of cells from *Msi2*<sup>eGFP/+</sup>; *Kras*<sup>G12D/+</sup>; *Trp53*<sup>fl/fl</sup>; *Rosa26*<sup>CreERT2/+</sup> tumorspheres, quaternary passage tumorspheres were dissociated from surrounding Matrigel in the manner described. Cells were resuspended in full media, mixed at a 1:1 ratio with growth factor reduced Matrigel (Corning) at a final volume of 100 µl, and transplanted subcutaneously into the flanks of NSG recipient mice. Tumor growth was monitored 1–2× a week via caliper measurement for 13 weeks and did not exceed 2 cm in diameter. At endpoint, both the flank tumor and lungs were collected for fixation and H&E analysis in the manner described. Lung H&E sections were analyzed for the presence of metastatic tumors.

## shRNA lentivirus production

For knockdown of Msi2, shRNA constructs and scrambled control were designed as previously described (*Fox et al., 2016*). For knockdown of Ptgds, Arl2bp, Rnf157, and Syt11, shRNA constructs and scrambled control were designed as previously described (*Lytle et al., 2019*) using the following target sequences: For Ptgds, 5′-ACCTCTACCTTCCTCAGGAAA-3′; for Arl2bp, 5′-GCTGCTCACATT CACGGATTT-3′; for Rnf157, 5′- CAGAGGGAAATGATATCATAG-3′; for Syt11, 5′-ATCAGGCTTCTC TGGGTTATT-3′.

## qRT-PCR analysis

RNA isolation was performed using RNeasy Micro and Mini kits (QIAGEN) and cDNA conversion was performed using Superscript III (Invitrogen). Quantitative real-time PCR was performed using an iCycler (Bio-Rad) in a mix containing iQ SYBR Green Supermix (Bio-Rad), cDNA, and gene-specific primers. Primer sequences for target genes are as follows: Msi2: forward 5′-TGCCATACACCATGGA TGCGT-3′/reverse 5′-GTAGCCTCTGCCATAGGTTGC-3′; Ptgds: 5′-GAAGGCGGCCTCAATCTCA-3′/ reverse 5′-CGTACTCGTCATAGTTGGCCTC-3′; Arl2bp: 5′-ACGATGGACGCCCTAGAAGA-3′/reverse 5′-GCAATAACTGGAACTCATCATCCAT-3′; Rnf157: 5′-CGCATTTCTAGGCATGGAGTG-3′/reverse 5′-GCCAGGTACCGGGTAAGC-3′; Syt11: 5′-GAGATCACAAATATACGCCCCAG-3′/reverse 5′-GCAG CACGTCCACACAAAG-3′; B2M: 5′-ACCGGCCTGTATGCTATCCAGAA-3′/reverse 5′-AATGTGAG GCGGGTGGAACTGT-3′. All real-time data were normalized to B2M.

## RNAseq and bioinformatics analysis

For RNAseq of lung epithelial cells from *Msi2*<sup>eGFP/+</sup>; *Kras*<sup>G12D/+</sup>; *Trp53*<sup>fl/fl</sup>; *Rosa26*<sup>CreERT2/+</sup> mice, 60,000 cells were isolated for the Msi2+ (Lin−EpCAM+GFP+) and Msi2− (Lin−EpCAM+GFP) populations via FACS in the manner described. For RNAseq of Msi2 knockdown in KP cells, KP cells were transduced with either shControl or shMsi2 lentivirus in triplicate for each group, and ≥125,000 transduced cells were isolated via sorting. RNA was isolated using a RNeasy Micro Kit (QIAGEN). The quality of total RNA was assessed using an Agilent Tapestation and all samples had an RIN ≥8. M2KD RNA libraries were generated from 2 µg of RNA, and *Msi2*<sup>eGFP/+</sup>; *Kras*<sup>G12D/+</sup>; *Trp53*<sup>fl/fl</sup>; *Rosa26*<sup>CreERT2/+</sup> libraries were generated from 90 ng of RNA using Illumina's TruSeq Stranded mRNA Sample Prep Kit following the manufacturer's instructions. RNA libraries were multiplexed and sequenced with 50 basepair (bp) single end reads (SR50) to a depth of approximately 30 million reads per sample on an Illumina HiSeq4000.

RNAseq and cell state analyses were performed as previously described (*Lytle et al., 2019*). For validation of differential expression analysis, edgeR and sleuth were used. Briefly, RNAseq fastq files were processed within the systemPipeR pipeline (*H Backman and Girke, 2016*). Alignment of the reads was performed using HISAT2 with the mm10 mouse assembly using default settings (*Kim et al., 2019*). Read counting was performed with the summarize Overlaps R packages with Union mode and the edgeR package was used for differential expression analysis using a fold change >1.9 and an FDR (False discovery rate) <0.05 (*Lawrence et al., 2013*; *Robinson et al., 2010*). As a secondary validation, transcript quantifications were performed using kallisto and sleuth was used for gene-level quantifications and differential expression analysis using default setting (*Bray et al., 2016*; *Pimentel et al., 2017*).

## Statistical analysis

Statistical analysis was performed using GraphPad Prism software version 7.0 (GraphPad Software Inc). Data are mean ± SEM. Two-tailed unpaired Student's $t$-tests and log-rank tests were used to determine statistical significance. The Grubb's test was used to identify and remove statistical outliers where indicated.

## Acknowledgements

We are grateful to Reuben J Shaw for scientific advice; Laurie Gerkin and Amanda Hutchins for technical training; Lillian J Eichner for technical advice; Marcie Kritzik for assistance with preparation of manuscript and figures; and Mathew Chvasta for assistance with figures.

## Additional information

### Funding

| Funder | Grant reference number | Author |
|---|---|---|
| Ruth L. Kirschstein NRSA Cancer Therapeutics Training Program | | Alison G Barber |
| Tobacco-Related Disease Research Program | | Alison G Barber Michael Hamilton Nirakar Rajbhandari |
| National Institutes of Health | T32 HL086344 | Cynthia M Quintero Michael Hamilton |
| National Institutes of Health | R35 CA197699 | Tannishtha Reya |

The funders had no role in study design, data collection and interpretation, or the decision to submit the work for publication.

### Author contributions

Alison G Barber, Conceptualization, Data curation, Formal analysis, Funding acquisition, Investigation, Methodology, Writing – original draft, Writing – review and editing; Cynthia M Quintero, Data curation, Formal analysis, Funding acquisition, Investigation, Writing – original draft, Writing – review and editing; Michael Hamilton, Formal analysis, Funding acquisition; Nirakar Rajbhandari, Data curation, Funding acquisition, Investigation; Roman Sasik, Formal analysis; Yan Zhang, provided scientific advice; Carla Kim, provided scientific advice; Hatim Husain, Resources, provided scientific advice; Xin Sun, performed data analysis and provided scientific advice; Tannishtha Reya, Conceptualization, Resources, Formal analysis, Supervision, Funding acquisition, Writing – original draft, Writing – review and editing

### Author ORCIDs

Tannishtha Reya ⓘ https://orcid.org/0000-0002-5956-8536

### Ethics

All animal experiments were performed according to protocols approved by the University of California San Diego Institutional Animal Care and Use Committee (#S12100).

Reviewer #1 (Public Review): https://doi.org/10.7554/eLife.97021.2.sa1
Reviewer #2 (Public Review): https://doi.org/10.7554/eLife.97021.2.sa2
Reviewer #3 (Public Review): https://doi.org/10.7554/eLife.97021.2.sa3
Author response https://doi.org/10.7554/eLife.97021.2.sa4

## Additional files

### Supplementary files
MDAR checklist

### Data availability
RNAseq datasets have been deposited into the NCBI BioProject database under the BioProject ID number PRJNA1195961.

The following dataset was generated:

| Author(s) | Year | Dataset title | Dataset URL | Database and Identifier |
|---|---|---|---|---|
| Reya et al. | 2024 | Effects of Musashi2 Knockdown in Lung Cancer | https://www.ncbi.nlm.nih.gov/bioproject/PRJNA1195961 | NCBI BioProject, PRJNA1195961 |

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
