## [Editor Report · eLife assessment]

This **important** study shows a significant role for Musashi-2 (Msi2) in lung adenocarcinoma. The authors provided **solid** data that support the requirement for Msi2 in tumor growth and progression, although the study would have been strengthened by including more patient samples and additional evidence regarding Msi2+ cells being more responsive to transformation. These findings are of interest to both the lung cancer and the RNA binding protein fields.

---

## [Referee Report · Reviewer #1 (Public Review)]

Summary:

Here, the authors, Barber AG et al, developed a new mouse model and investigated an importance of Musashi-2 in lung cancer. Specifically, they found that Musashi-2 is important for lung cancer cells as it controls cancer cell growth, and also regulates several genes that also control cancer cell growth. Development of a new Musashi-2 mouse model is a plus, which confirmed Musashi-2 importance for lung cancer survival, and finding several genes that Musashi controls that are important for lung cancer growth. Additionally, they demonstrated that Musashi-2 overexpression which is tracked by GFP is preferred in lung adenocarcinoma cells. The data is rigorous and only minor revisions are requested.

Strengths:

Authors achieved their goals, by developing new Musashi-2 mouse model, confirming Musashi-2 importance for lung cancer survival, and finding several genes that Musashi controls that are important for lung cancer growth.

Weaknesses:

The findings of Musashi-2 mouse and human lung cancer growth control are not that novel as prior publication in 2016 showed that already, again, in both human and mouse models (Kudinov et al PNAS, PMID: 27274057), and also the authors missed the point of that paper which did use both miuse and human models to show impact on inbvasion and metastasis- both in vitro and in vivo. Additionally, another publication is currently under revisions recently also generated new Musashi-2 transgenic mouse model which confirmed Musashi-2 support of lung cancer growth (Bychkov I et al, PMID: 37398283; https://www.biorxiv.org/content/10.1101/2023.06.13.544756v1). Another weakness is that Musashi-2 cannot be effectively targeted and the new genes the authors found that Musashi-2 regulates are likely to be also difficult therapeutic targets. Therefore, impact of this new investigation is relatively modest in the field.

Major suggestions:

(1) Figure 3: it is unclear what is the efficiency of Msi2 deletion shRNA - could you demonstrate it by at least two independent methods? (QPCR, Western, or IHC?) please quantitate the data.

(2) In Figure 4, similarly, it is unclear if Msi2 depletion was effective- and what is shRNA efficiency. Please test this by at least two independent methods (QPCR, Western, or IHC) and also please quantitate the data

(3) the reason for impairment of cell growth demonstrated in Figs 3 and 4 is not clear: is it apoptosis? Necrosis? Cell cycle defects? Autophagy? Senescence? Please probe 2-3 possibilities and provide the data.

(4) Since Musashi-1 is a Musashi-2 paralogue that could compensate for Musashi-2 loss, please test Msi1 expression levels in matching Fig 3 and Fig 4 sections (in cells/ tumors with Msi2 deletion and in KP cells with Msi2 shRNA). One method could suffice here.

(5) It is not exactly clear why RNA-seq (as opposed to proteomics) was done to investigate downstream Msi2 targets (since Msi2 is in first place, translational and not transcriptional regulator)- . RNA effects in Fig 5J are quite modest, 2-fold or so. It would be useful (if antibodies available) to test four targets in Fig 5J by Western blot, to see any impact of musashi-2 depletion on those target protein levels. Indeed, several papers - including Kudinov et al PNAS, PMID: 27274057, Makhov P et al PMID: 33723247 and PMID: 37173995 - used proteomics/ RIP approaches and found direct Musashi-2 targets in lung cancer, including EGFR, and others.

---

## [Referee Report · Reviewer #2 (Public Review)]

Summary:

Alison G. Barber et al. reports the function of Msi2 in mouse models of non-small cell lung cancer. The expression of Msi2 in normal lung was evaluated using a knockin reporter allele. Msi2 expressing cells were found to be around 30-40% in normal lung epithelium without a strong bias in subsets of lung cells. Knocking out Msi2 in a KrasG12D and P53 KO model reduced lung cancer initiation. Knocking down Msi2 in established lung cancer cells reduced in vitro sphere formation and in vivo xenograft. Finally, the authors identified several genes whose expression was downregulated by Msi2 knockdown. Knocking down four of these genes, including Ptgds, Arl2bp, hRnf157, and Syt11, each with a single shRNA, reduced lung sphere formation in vitro, suggesting their involvement in lung cancer.

Strengths:

This manuscript represents an interesting advance on the role of Msi2 in lung cancer. While some of the data (for example the knockdown effect of Msi2 in established lung cancer cells) corroborated previous findings, the study of Msi2 expression in normal lung and the characterization of the KO phenotype in lung cancer initiation are new and interesting.

Weaknesses:

Two areas can be further strengthened. Several conclusions are not fully supported by the existing data. The stable/dynamic nature of Msi2 expressing cells in lung would benefit from more detailed investigations for proper data interpretation.

(1) It will be interesting to determine whether Msi2+ cells are a relatively stable subset or rather the Msi2+ cells in lung is a dynamic concept that is transient or interconvertible. This is relevant to the interpretation of what Msi2 positivity really means.

(2) Does Kras mutation and/or p53 loss upregulate Msi2? This point and the point above are related to whether Msi2+ cells are truly more susceptible to tumorigenesis, as the authors suggested.

(3) The KO of Msi2 reducing tumor number and burden in the lung cancer initiation model is interesting. However, there are two alternative interpretations. First, it is possible that the Msi2 KO mice (without Kras activation and p53 loss) has reduced total lung cell numbers or altered percentage of stem cells. There is currently only one sentence citing data not shown on line 125, commenting that there is no difference in BASC and AT2 cell populations. It will be helpful that such data are shown and the effect of KO on overall lung mass or cellularity is clarified. Second, the phenotype may also be due to a difference in the efficiencies of cre on Kras and p53 in the Msi2 WT and KO mice.

(4) All shRNA experiments (for both Msi2 KD and the KD of candidate genes) utilized a single shRNA. This approach cannot exclude off-target effects of the shRNA.

(5) The technical details of the PDX experiment (Figure 4F) are not fully explained.

---

## [Referee Report · Reviewer #3 (Public Review)]

Summary:

In this manuscript, Barber and colleagues propose a dual role for the RNA-binding protein Mushashi-2 (Msi2) in lung adenocarcinoma initial transformation and subsequent tumor propagation. First, authors show that Msi2 is expressed in a subset of Club/BASC (37%) and AT2 (26%) cells in the normal lung and displayed a distinct transcriptional profile than non-expressing Msi2 cells. Furthermore, Msi2 is broadly expressed/activated in vivo in genetically induced lung adenocarcinoma tumors (Kras/p53 mouse model) and Msi2+ cells displayed a significantly higher ability to form tumor spheres in vitro. Authors demonstrated by in vivo and in vitro assays that Msi2 loss of function significantly impair tumor growth and progression in lung adenocarcinoma. Data showed that Msi2 function is conserved in human adenocarcinoma tumor growth in patient-derived xenograft. Lastly, novel genes regulated by Msi2 and involved in lung adenocarcinoma tumor growth were identified.

Strengths:

The authors provided convincing data for a key role of Msi2 in lung adenocarcinoma tumor progression and growth. Multiple evidences using Msi2 knock-out genetic mouse model and shRNA knock-down in tumor sphere formation assay are clearly demonstrated. The conservation and importance of Msi2 was further shown in human patient-derived xenograft. Although specific cell types (Club/BASC, AT2) were not isolated, authors further delved in the transcriptional difference between Msi2+ and Msi2- cells in the normal lung. Furthermore, novel genes and pathways regulated by Msi2 in lung adenocarcinoma were identified and tested for their ability to inhibit tumor growth in vitro. These 2 RNA-Seq datasets will be useful in the future and provide a basis to explore (1) potential propensity of a given cell to initiate oncogenic transformation, and (2) potential novel regulators of lung adenocarcinoma.

Weaknesses:

Although this work strongly demonstrated the importance of Msi2 in lung adenocarcinoma tumor progression and growth, the following points remain to be clarified or addressed.

- In Figure 1, characterization of Msi2 expression in the normal mouse lung was carried out by using a Msi2-GFP Knock-in reporter and analyzed by flow cytometry followed by cytospins and immunostaining. Additional characterization of Msi2 expression by co-immunostaining with well-known markers of airway and alveolar cell types in intact lung tissue will strengthen the existing data and provide more specific information about Msi2 expression and abundancy in relevant cell types. It will be also interesting to know whether Msi2 is expressed or not in other abundant lung cell types such as ciliated and AT1 cells.

- While this set of experiments provide strong evidence that Msi2 is required for tumor progression and growth in lung adenocarcinoma, it is unclear whether normal Msi2+ lung cells are more responsive to transformation or whether Msi2 is upregulated early during the process of tumorigenesis. Future lineage tracing experiments using Msi2-CreER and mouse models of chemically-induced lung carcinogenesis will provide additional data that will fully support this claim.

- In Figure 4F, Patient-derived xenograft (PDX) assays were conducted in 2 patients only and the percentage of cells infected by shRNA-Msi2 is low in both PDX (30% and 10% for patient 1 and 2 respectively). It is surprising that Msi2 downregulation in a small percentage of tumor cells has such a dramatic effect on tumor growth and expansion. Confirmation of this finding with additional patient samples would suggest an important non-cell autonomous role for Msi2 in lung adenocarcinoma.

---

## [Author Response]

**Reviewer #1 (Public Review):**
(1) Figure 3: it is unclear what is the efficiency of Msi2 deletion shRNA - could you demonstrate it by at least two independent methods? (QPCR, Western, or IHC?) please quantitate the data.

In Figure 3, we did not delete Msi2 via shRNA. Instead, we utilized a genetic model in which the Msi2 gene was disrupted via gene trap mutagenesis. We have also used this model in previous publications to define the impact of Msi2 loss in other systems1.

(2) In Figure 4, similarly, it is unclear if Msi2 depletion was effective- and what is shRNA efficiency. Please test this by at least two independent methods (QPCR, Western, or IHC) and also please quantitate the data

We demonstrated that the efficiency of Msi2 depletion was ~83% (Figures 4A and 4C) via qPCR analysis for our in vitro and in vivo experiments, respectively, and verified the knockdown via bulk RNA-seq analysis. The shRNA hairpin used was previously validated and published by our lab2.

(3) the reason for impairment of cell growth demonstrated in Figs 3 and 4 is not clear: is it apoptosis? Necrosis? Cell cycle defects? Autophagy? Senescence? Please probe 2-3 possibilities and provide the data.

The basis of the cell growth impairment after Msi2 deletion/knockdown in this paper is certainly an important question, and future experiments will be performed to better delineate this. In previous publications loss of Msi2 in leukemia cells has been shown to inhibit growth via arrested cell cycle progression by increasing the expression of p213. Further, loss of Msi2 was also shown to promote apoptosis in part by upregulating Bax3. These data suggest that Msi2 can have an impact via multiple distinct mechanisms including by mediating cell cycle arrest and blocking apoptosis. While these specific genes were not detectably changed after loss of Msi2 in lung cancer cells, other genes in these and other pathways will be important to study in the future.

(4) Since Musashi-1 is a Musashi-2 paralogue that could compensate for Musashi-2 loss, please test Msi1 expression levels in matching Fig 3 and Fig 4 sections (in cells/ tumors with Msi2 deletion and in KP cells with Msi2 shRNA). One method could suffice here.

In our RNA-seq of cells following Msi2 knockdown, Msi1 expression was undetectable. The TPM values for Msi1 in control and knockdown cells were less than 0.01, suggesting that it did not compensate for the loss of Msi2.

(5) It is not exactly clear why RNA-seq (as opposed to proteomics) was done to investigate downstream Msi2 targets (since Msi2 is in first place, translational and not transcriptional regulator)- . RNA effects in Fig 5J are quite modest, 2-fold or so. It would be useful (if antibodies available) to test four targets in Fig 5J by Western blot, to see any impact of musashi-2 depletion on those target protein levels. Indeed, several papers - including Kudinov et al PNAS, PMID: 27274057, Makhov P et al PMID: 33723247 and PMID: 37173995 - used proteomics/ RIP approaches and found direct Musashi-2 targets in lung cancer, including EGFR, and others.

Previous published work from the lab showed that expression of Msi2 in the context of myeloid leukemia1can not only repress NUMB protein (I believe protein should be all caps?) (as has been previously demonstrated in the nervous system) but also Numb RNA. This indicated that as an RNA binding protein, Msi2 also can bind and destabilize direct binding targets such as Numb; this was the reason for pursuing transcriptomic analysis. However as the reviewer suggests, proteomic studies are certainly very important to develop a complete picture of the impact of Musashi to determine which targets are controlled by Msi2 at the protein level.

**Reviewer #2 (Public Review):**
(1) It will be interesting to determine whether Msi2+ cells are a relatively stable subset or rather the Msi2+ cells in lung is a dynamic concept that is transient or interconvertible. This is relevant to the interpretation of what Msi2 positivity really means.

In previous unpublished work from our lab, we have found that Msi2+ cells from a GFP reporter KPf/fC mouse are readily able to become GFP negative (Msi2-), but the inverse is not true. Specifically, when Msi2+ KPf/fC pancreatic cells were transplanted into the flanks of NSG mice, Msi2+ cells formed tumors in all recipients; these tumors contained both GFP+ and GFP- cells (over 80%) recapitulating the original heterogeneity and suggesting GFP+ cells can give rise to both GFP+ and GFP- cells (Lytle and Reya, unpublished observations). In contrast only a small subset of GFP- transplanted mice formed tumors. One of the rare GFP- derived tumors was isolated and found to contain largely GFP- cells, with ~0.1% GFP+ cells. The small frequency of GFP expression could be from contaminating cells or may suggest that GFP- cells retain some ability to switch on Msi under selective pressure, and that although they pose a lower risk of driving tumorigenesis than Msi+ cells, they may nonetheless bear latent potential to become higher risk. These data may offer a possible model for projecting the potential of Msi2+ cells in the lung, but is something that needs to be further studied in this tissue.

(2) Does Kras mutation and/or p53 loss upregulate Msi2? This point and the point above are related to whether Msi2+ cells are truly more susceptible to tumorigenesis, as the authors suggested.

In unpublished work from our lab, we have found that Kras mutation upregulates Msi2 over baseline and subsequent p53 loss upregulates Msi2 further in the context of pancreatic cells (Lytle and Reya unpublished results), therefore it is possible that the same is true for the lung. Specifically, we have observed that Msi2 increased from normal acinar cells to Kras-mutated acinar (e.g. pancreatic intraepithelial neoplasia (PanIN)).

To address whether Msi2+ cells are more susceptible to tumorigenesis, we have recently published data showing that the stabilization of the oncogenic MYC protein in lung Msi2+ cells drive the formation of small-cell lung cancer in a new inducible Msi2-CreERT2; CAG-LSL-MycT58A mice (Msi2-Myc)4 model. More importantly, this data provides the first evidence that normal Msi2+ cells are primed and highly sensitive to MYC-driven transformation across many organs and not just the lung4.

(3) The KO of Msi2 reducing tumor number and burden in the lung cancer initiation model is interesting. However, there are two alternative interpretations. First, it is possible that the Msi2 KO mice (without Kras activation and p53 loss) has reduced total lung cell numbers or altered percentage of stem cells. There is currently only one sentence citing data not shown on line 125, commenting that there is no difference in BASC and AT2 cell populations. It will be helpful that such data are shown and the effect of KO on overall lung mass or cellularity is clarified. Second, the phenotype may also be due to a difference in the efficiencies of cre on Kras and p53 in the Msi2 WT and KO mice.

We isolated the lungs of three Msi2 WT and three Msi2 KO mice and used immunofluorescence staining to stain for CC10 (BASC) and SPC (AT2) to determine if these cell populations were reduced after Msi2 loss alone. Below are representative images showing that the Msi2 KO mice did not have lower numbers of both BASC and AT2 cell populations.

**Author response image 1. sa4fig1:** 

(4) All shRNA experiments (for both Msi2 KD and the KD of candidate genes) utilized a single shRNA. This approach cannot exclude off-target effects of the shRNA.

The shRNA hairpin used for Msi2 was previously validated and published by our lab2. Additionally, in this work we did develop and use a Msi2 genetic knockout mouse model that validates our shRNA knockdown data showing the specific impact of Msi2 on lung tumor growth.

(5) The technical details of the PDX experiment (Figure 4F) are not fully explained.

Due to space considerations, we were unable not put the specifics in the legend, but the details are in the methods section (Flank Transplant Assays). In brief, 500,000 cells/well were plated in a 6-well plate coated with Matrigel and 83,000 cells/well were plated in a 24-well plate coated with Matrigel for subsequent determination of transduction efficiency via FACS. 24 hours after transduction, media from the cells was collected and placed on ice. 1mL of 2mg/mL collagenase/dispase was then added to the well and incubated for 45 minutes at 37ºC to dissociate the remaining cells from Matrigel followed by subsequent washes. Cells were pelleted by centrifugation and an equivalent number of shControl and shMsi2 transduced cells were resuspended in full media, mixed at a 1:1 ratio with growth factor reduced Matrigel at a final volume of 100 μL, and transplanted subcutaneously into the flanks of NSG recipient mice.

**Reviewer #3 (Public Review):**
- In Figure 1, characterization of Msi2 expression in the normal mouse lung was carried out by using a Msi2-GFP Knock-in reporter and analyzed by flow cytometry followed by cytospins and immunostaining. Additional characterization of Msi2 expression by co-immunostaining with well-known markers of airway and alveolar cell types in intact lung tissue will strengthen the existing data and provide more specific information about Msi2 expression and abundancy in relevant cell types. It will be also interesting to know whether Msi2 is expressed or not in other abundant lung cell types such as ciliated and AT1 cells.

We performed co-staining of Msi2 and CC10 as well as Msi2 and SPC in Figure 1C. In the future we can include additional markers as well as markers for airway and other alveolar cell types.

- While this set of experiments provide strong evidence that Msi2 is required for tumor progression and growth in lung adenocarcinoma, it is unclear whether normal Msi2+ lung cells are more responsive to transformation or whether Msi2 is upregulated early during the process of tumorigenesis. Future lineage tracing experiments using Msi2-CreER and mouse models of chemically-induced lung carcinogenesis will provide additional data that will fully support this claim.

Recently, we published data showing that Msi2 is expressed in Clara cells at the bronchoalveolar junction in the lung of our new Msi2-CreERT2 knock-in mouse model4. Furthermore, stabilization of the oncogenic MYC protein in these specific cells to model Myc amplification was sufficient to drive the formation of small-cell lung cancer4. These data excitingly demonstrate that Msi2+ cells are more responsive to transformation after Myc stabilization.

- In Figure 4F, Patient-derived xenograft (PDX) assays were conducted in 2 patients only and the percentage of cells infected by shRNA-Msi2 is low in both PDX (30% and 10% for patient 1 and 2 respectively). It is surprising that Msi2 downregulation in a small percentage of tumor cells has such a dramatic effect on tumor growth and expansion. Confirmation of this finding with additional patient samples would suggest an important non-cell autonomous role for Msi2 in lung adenocarcinoma.

In the future we hope to collect more patient samples to further validate the data presented with the first 2 patients shown here. We are not certain about the reason behind the large impact of Msi2 inhibition, but as cancer stem cells drive the formation of the rest of the tumor and also drive the stromal microenvironment, it is possible that when Msi2 is deleted, Msi2- cells no longer form tumors? and also the ability to build the stromal microenvironment is impacted. This possibility needs to be further tested in future experiments.

**References**

(1) Ito, T. Kwon, H. Y., Zimdahl, B., Congdon, K. L., Blum, J., Lento, W. E., Zhao, C., Lagoo, A., Gerrard, G., Foroni, L., Goldman, J., Goh, H., Kim, S. H., Kim, D. W., Chuah, C., Oehler, V. G., Radich, J. P., Jordan, C. T., & Reya, T. Regulation of myeloid leukaemia by the cell-fate determinant Musashi. Nature 466, 765–768 (2010).

(2) Fox, R. G. Lytle, N. K., Jaquish, D. V., Park, F. D., Ito, T., Bajaj, J., Koechlein, C. S., Zimdahl, B., Yano, M., Kopp, J. L., Kritzik, M., Sicklick, J. K., Sander, M., Grandgenett, P. M., Hollingsworth, M. A., Shibata, S., Pizzo, D., Valasek, M. A., Sasik, R., Scadeng, M., Okano, H., Kim, Y., MacLeod, A. R., Lowy, A. M., & Reya, T. Image-based detection and targeting of therapy resistance in pancreatic adenocarcinoma. Nature 534, 407–411 (2016).

(3) Zhang, H. Tan, S., Wang, J., Chen, S., Quan, J., Xian, J., Zhang, Ss., He, J., & Zhang, L. Musashi2 modulates K562 leukemic cell proliferation and apoptosis involving the MAPK pathway. Exp Cell Res 320, 119-27 (2014).

(4) Rajbhandari, N., Hamilton, M., Quintero, C.M., Ferguson, L.P., Fox, R., Schürch, C.M., Wang, J., Nakamura, M., Lytle, N.K., McDermott, M., Diaz, E., Pettit, H., Kritzik, M., Han, H., Cridebring, D., Wen, K.W., Tsai, S., Goggins, M.G., Lowy, A.M., Wechsler-Reya, R.J., Von Hoff, D.D., Newman, A.M., & Reya, T. Single-cell mapping identifies MSI+ cells as a common origin for diverse subtypes of pancreatic cancer. Cancer Cell 41(11):1989-2005.e9 (2023).